# Chemometric Tools Associated with Quality Parameters for Evaluation of *Mauritia flexuosa* L.f. Oil in the State of Pará (Brazil)

**DOI:** 10.3390/foods14091585

**Published:** 2025-04-30

**Authors:** Braian Saimon Frota da Silva, Nelson Rosa Ferreira, Fábio Dos Santos Gil, Simone de Fátima Pinheiro Pereira, Alana Coêlho Maciel, Claúdio Nahum Alves

**Affiliations:** 1Graduate Programs in Chemistry (PPGQ), Federal University of Pará (UFPA), Belém 66075-110, Pará, Brazil; fabiogil10@yahoo.com.br (F.D.S.G.); simonefpp@gmail.com (S.d.F.P.P.); nahum@ufpa.br (C.N.A.); 2Graduate Program in Food Science and Technology (PPGCTA), Institute of Technology (ITEC), Federal University of Pará (UFPA), Belém 66075-110, Pará, Brazil; nelsonrosa@ufpa.br (N.R.F.); alanacoelhomaciel1@gmail.com (A.C.M.)

**Keywords:** vegetable oil, quality parameters, carotenoids, PCA, ANOVA

## Abstract

*Mauritia flexuosa* L.f. oil has high added value due to its antioxidant activity. This study evaluated the quality of 50 samples from eight regions of Pará, using analytical and chemometric methods. Total carotenoids, polyphenols, flavonoids, vitamin C, acidity, peroxide, saponification, and fatty acid indices were analyzed. Samples with higher residual load were identified by the Mahalanobis distance. Principal component analysis (PCA) highlighted total carotenoids as the most relevant parameter, identifying three groups with different levels of biological activity (low, medium, and high) and significant cumulative variance. ANOVA did not indicate significant differences between groups regarding fatty acids, but oleic acid was predominant in five regions. Nine samples are suitable for the food sector, while the others can be directed to different applications. The maximum concentrations of carotenoids, polyphenols, flavonoids, and vitamin C were 1899 μg/g, 161.69 GAE/100 g, 125.02 mg EC/100 g, and 24.17 mg/100 g, respectively. This study demonstrated the usefulness of chemometric tools in the quality control of this bioinput, facing the local bioeconomy.

## 1. Introduction

The Legal Amazon has important biodiversity, with thousands of specimens of high biotechnological potential. Among them are vegetables, which contain natural oils rich in nutrients and several molecules of interest that provide health benefits [1]. In particular, buriti oil has several properties with distinct effects, such as, for example, maintaining the state of body homeostasis [2].

Buriti is also known as muriti, carandá-guaçú, carandaí-guaçu, palmeira-buriti, palmeira-dos-brejos, mariti, bariti, or meriti (*Mauritia flexuosa* L.f.). It is an angiosperm that has its origins in the Cerrado biome and the Amazon biome and contains in its nutritional composition a high content of β-carotene, flavonoids, α-tocopherol, saturated and unsaturated fatty acids, phenolic compounds, and ascorbic acid, among other compounds [3,4].

The fruits and oil of this vegetable can be economically viable in gastronomy, in the preparation of sweets, jellies, ice creams, food colorings, flour, and cookies [5,6]. Different extraction methods can significantly impact the amount of buriti oil obtained. By using the artisanal method, it is possible to achieve an average of 4%. However, higher rates are achieved by hydraulic pressing (21–45%) and solvent extraction (24%) [7,8]. Buriti oil has an average of 20% saturated fatty acids in its chemical composition, 76% monounsaturated, and 4% polyunsaturated fatty acids [9,10].

The compounds present in buriti oil, such as carotenoids, tocopherols, and phenolic compounds, have the potential to prevent diseases associated with oxidative stress [7,11]. As per the research conducted by Muscolo et al. [12], including carotenoid-rich foods in our diet can aid in preventing certain neurodegenerative diseases, cancer, and cardiovascular diseases.

Phenolic compounds possess therapeutic properties such as anti-inflammatory, antidepressant, antitumor, antilipid, diabetes control, and antimicrobial properties [13,14]. A study by Cruz et al. [11] has suggested that the bactericidal action in vegetable oils, including buriti, is due to phenolic compounds.

The quality of vegetable oils must be carefully evaluated, as there is still the possibility of adulteration to minimize processing costs and maintain a high commercial value. This action goes against the provisions of Resolution No. 270 of 22 September 2005 of the Health Surveillance Agency (ANVISA/BRAZIL), Codex Alimentarius, Food and Agriculture Organization of the United Nations (FAO) and the World Health Organization (WHO) [15,16,17].

For oils to be used for different commercial and industrial purposes, it is essential to define quality parameters such as acidity, saponification, and peroxide levels, as stated by ANVISA [15]. Additionally, the existence of bioactive compounds is also a significant quality parameter [18,19].

Chemical analyses can be time-consuming and labor-intensive despite their relative simplicity. To overcome these challenges, UV-vis spectrophotometry, when combined with chemometric methods, becomes a powerful tool for quantitative and qualitative analysis of analytes [20]. Chemometric methods improve the accuracy, efficiency, and analysis capability of spectrophotometric data, enabling the resolution of complex mixtures, development of robust calibration models, and automated processing of large volumes of data [21].

Principal Component Analysis (PCA) is a multivariate statistical technique used to reduce the dimensionality of data, keeping the most essential information. This is accomplished by converting the initial data into a new set of variables, linear combinations of the original variables [22]. When analyzing pure oil samples and samples suspected of adulteration, PCA can identify possible adulterations by examining the distribution of eigenvectors in the principal components.

This technique can also be used to group similar vegetable oil samples, detect anomalies in the data, and ensure the quality and authenticity of food products. Studies by Hosseini, Beheshti, Minaei [23], and Meng et al. [24] have shown that PCA can be a powerful tool for identifying significant differences in distributions of principal components, indicating potential adulterations. Similarly, studies by Ceniti et al. [25] and Vladić et al. [26] have demonstrated the usefulness of PCA in ensuring the quality and authenticity of food products.

Considering a direct application, PCA can be used to investigate oil quality parameters from different regions, making this a valuable strategy to understand how geographic factors can affect the composition and quality of vegetable oils [27,28]. Collecting data from oil samples and applying PCA can identify patterns of variation in quality parameters, such as acidity, peroxide index, and fatty acid profile, among others. This information can help the food and agricultural industry optimize production, identify high-quality cultivation areas, and monitor the authenticity of vegetable oils, ensuring consistent quality standards across different regions [29,30,31].

Another statistically valuable technique in the quality control of vegetable oils is the one-way analysis of variance (one-way ANOVA). This technique allows us to compare the averages of multiple groups of vegetable oil samples, making it possible to identify significant variations in the properties or composition of different batches or collection points in different locations [32,33,34].

This study aims to investigate the quality profile of buritis oils in eight municipalities in Pará (Brazil) in accordance with specific legislation, and quantify polyphenols, total carotenoids, ascorbic acids, flavonoids, and fatty acids. In addition, Principal Component Analysis and one-way ANOVA were applied to the set of parameters to understand the quality profile of the oils in each region. We emphasize that there is the possibility of later choosing other reference parameters to contribute information to the food and agricultural industries, among other economic sectors. This makes it possible to optimize production, identify high-quality cultivation areas, and monitor the authenticity of these oilseed matrices.

## 2. Materials and Methods

### 2.1. Obtaining Samples

Fruit collection was conducted in eight municipalities in Pará, totaling 50 samples of 1 kg of buriti. The municipalities were Bujaru (1°30′54″ S, 48°02′42″ W), code BT1; fourteen samples from the municipality of Bragança (01°03′40″ S, 46°45′16″ W), codes BT2 to BT2D; seventeen samples from the municipality of Viseu (1°11′49″ S, 46°8′24″ W), codes BT3 to BT3G; eight samples from the municipality of Igarapé-Miri (1°58′30″ S, 48°57′36″ W), codes BT4 to BT47; a sample from Ilha das Onças in the municipality of Barcarena (1°27′10″ S, 48°32′46″ W), code BT5; three samples from the municipality of Acará (1°57′39″ S, 48°11′49″ W), codes BT6 to BT62; five samples in the municipality of Belém (1°27′21″ S, 48°30′14″ W), codes BT7 to BT74; and one sample in the municipality of Santarém (2°26′34″ S, 54°42′28″ W), code BT8. After collection, the seeds were dried in an oven (model Inova 220v, Votorantim, São Paulo, Brazil) at a temperature of 45 °C for six hours. Next, the seeds were pressed at room temperature (25 °C) using a mechanical press (ERT 60, from Scoot Tech, Vinhedo, São Paulo, Brazil). The oils obtained were refrigerated in opaque containers to preserve their quality.

### 2.2. Acid Index

The acid index (AI) represents the amount in milligrams of KOH necessary to neutralize 1 g of the natural oil sample, according to the American Oil Chemists’ Society—AOCS, Cd 3d-63 [35].

### 2.3. Peroxide Index

Peroxides and hydroperoxides are primary products of lipid oxidation resulting from the breakdown of unsaturated fatty acid chains. According to AOCS, Cd 8b-90, the peroxide index (PI) represents the amount of oxygen peroxide per kilogram of oil [36].

### 2.4. Saponification Index

The saponification index expresses the amount in mg of KOH per gram of oil in alkaline hydrolysis. The saponification index (SI) was determined according to the AOCS Cd 3-25 methodology [37].

### 2.5. Total Carotenoid Content

Sequential dilutions of 0.15 g of the extracted oil were initially prepared in 10 mL of petroleum ether. Subsequently, readings were taken on a spectrophotometer at 450 nm to protect the light action. The procedure was performed in triplicate. The concentration of total carotenoids was calculated in micrograms of carotenoids per gram of oil (μg/g) using the equation proposed by Davies [38]:(1)$$Totalcarotenoids=A\times V\times \frac{10^{4}}{A_{1cm}^{1\%}}$$
where the variables are as follows: A (absorbance at 450 nm); V (solution volume); and $A_{1cm}^{1\%}$ (absorption coefficient of β-carotene in petroleum ether = 2592) [39].

### 2.6. Total Phenolic Content

The total phenolic content of buriti oil was determined according to the Folin–Ciocalteu method, described by Ramos-Escudero et al. [40] with adaptations. The 2 g sample of oil was diluted in 5 mL of hexane and 5 mL of methanol, centrifuged for 1 h at 2.08 Hz, and then filtered. Afterwards, 1 mL of the filtrate was mixed with 2.5 mL of Folin–Ciocalteu reagent (10%) and centrifuged for 5 min at 50 Hz. The recovered polar fraction was mixed with 2 mL (7.5% *w*/*v*) of sodium carbonate and allowed to react for 1 h at 37 °C in a water bath. The blue color formed was measured at 760 nm (EVEN UVM90 spectrophotometer—Bel Engineering, Monza, MB, Italy). The results were expressed in milligram gallic acid equivalents per 100 g of oil.

### 2.7. Total Flavonoid Content

The method used to determine the total flavonoid content in buriti oil was based on the Best et al. [41] protocol with some adjustments. Initially, 2 g of buriti oil was diluted in 10 mL of ethanol and centrifuged for an hour at 2.08 Hz. The resulting supernatant was filtered and centrifuged again at 50 Hz for 20 min at a temperature of 20 °C. A 100-microliter sample of the upper phase was mixed with 75 microliters of a 5% (*w*/*v*) NaNO_2_ solution and allowed to rest for 5 min. Next, 150 microliters of a 10% (*w*/*v*) AlCl_3_·6H_2_O solution were added and left to rest for another 5 min. Finally, 500 microliters of a 1M NaOH solution were added, and the mixture was allowed to rest for 15 min. Absorbance was measured at 425 nm using an Even UVM90 spectrophotometer (Bel Engineering, Monza, MB, Italy). The total flavonoid content was expressed as milligrams of catechin equivalents per 100 g of the sample.

### 2.8. Ascorbic Acid Content

The method for determining the vitamin C content in buriti oil samples was followed as described by Oliveira et al. [42] with some modifications. Initially, 2.5 g of oil was diluted in 10 mL of 0.4% (*w*/*v*) oxalic acid solution and stirred gently for 2 h. The mixture was then vacuum-filtered, and the resulting solution was further diluted at a ratio of 1:10. Next, 1 mL of the diluted solution was added to 9 mL of dichloroindophenol solution. The mixture was centrifuged at 50 Hz for 5 min and left to rest for 10 min. Finally, the material was analyzed at 520 nm on an Even UVM90 UV-visible spectrometer (Bel Engineering, Monza, MB, Italy). The vitamin C content was expressed as mg of ascorbic acid per 100 g sample (mg/100 g).

### 2.9. Fatty Acid Profile

The acid profile was determined using a gas chromatograph (Shimadzu, Kyoto, Japan), model GC-2010, per the AOCS Ce 1a-13 standard [43]. The samples were prepared according to the AOCS Ce 2-66 standard [44] for methyl esters. The specifications used for injection were as follows: TG-WAX column 30.0 m × 0.32 mm; helium was used as the carrier gas at a flow rate of 1 mL/min; the injection volume was 1 μL (split, partition 1:10); the temperature ramp used was 1 min at 50 °C, followed by heating to 250 °C, where the sample was injected at a rate of 10 °C/min, and the temperature was held for 8 min. The peaks were analyzed using LabSolutions software (Shimadzu, Kyoto, Japan).

### 2.10. Statistical Analysis

Principal component analysis was applied to explore the reduced dimension and better visualize the experimental data of IA, IP, IS, carotenoids, polyphenols, flavonoids, and ascorbic acid, in addition to stipulating the biological activities. The preprocessing used was autoscaling in the management of the models. A value of *p* < 0.05 was considered to indicate a statistically significant difference. The Mahalanobis distance was applied to verify marked discrepancies between samples to adjust the model for efficient cumulative variance. Score and loading plots were generated from principal components (PCs) by MINITAB software version 14.13 (OSB Software, Bela Vista, SP, Brazil).

A one-way ANOVA statistical analysis was used to determine if there were significant differences in the average fatty acid levels among different regions of Pará. This analysis was performed in triplicate at a significance level of 95%, with a *p*-value of less than 0.05, followed by the Tukey test. The “box plot” graphics were generated using MINITAB software version 14.13. (OSB Software, Bela Vista, SP, Brazil).

## 3. Results and Discussion

### 3.1. Acid Index

The cold-pressed oils had a yield of approximately 20%. The acidity index (AI) results of 50 buriti oil samples are shown in Figure 1 with a population standard deviation of ±2.72. The sample with the lowest AI rating was BT73, collected from the municipality of Belém, with an AI rating of 0.80 mg of KOH/g. A low index of acidity in vegetable oils indicates better sensory properties such as flavor, odor, and stability, making it perfect for use in gastronomy as a high-quality gourmet oil [42,45,46]. On the other hand, oil samples from the municipality of Benevides, near Belém, had an average AI rating of 5.76 mg of KOH/g [47], which is seven times higher than the BT73 sample. Out of all the samples analyzed, only 33 (66%) had an AI value of less than 4 mg KOH/g, which complies with ANVISA Resolution N° 270, a technical regulation for cold-pressed and unrefined vegetable oils.

The AI value of oils measures the sensorial quality and protection of consumers’ health. The highest AI value observed was 19.17 mg of KOH/g of sample BT1, an oil extracted from the Bujaru region. When oils are exposed to adverse factors such as heat, light, oxygen, and humidity, the formation of free fatty acids occurs, which leads to an increase in the AI value [48,49].

Samples from BT73 to BT44 present AI values below 4 mg KOH/g. Samples with low acidity indexes can be used in the food industry and in the production of drugs that aim to prevent cardiovascular and neurodegenerative pathologies due to their potential antioxidant effect, as stated by De Andrade Mesquita et al. [50], Marcelino et al. [51], and Oliveira et al. [10]. On the other hand, the samples that have AI values above 4 mg KOH/g, specifically those from BT39 to BT1, can still be refined and used in other sectors of the economy, such as cosmetics and biofuels [52].

### 3.2. Peroxide Index

The peroxide index (PI) values are represented in Figure 2, with a population deviation of ±6.13, and indicate a variation ranging from 1.52 meq O_2_/kg (from the Bragança municipality sample, named BT25) to 19.64 meq O_2_/kg (from the Igarapé-Miri municipality sample). Of all the samples analyzed, only 30 (60%) samples had an IP below 15 meq O_2_/kg. The codes BT25 to BT3D represent different municipalities, including Bujaru (one sample), Bragança (thirteen samples), Viseu (six samples), Ilhas das Onças (one sample), Acará (three samples), Belém (five samples), and Santarém (one sample), all of which were at the quality limit for cold-pressed and unrefined oils [15,16]. In a study conducted by Serra et al. [48], samples were collected in the municipality of Benevides to determine the peroxide levels of buriti oil extracted using a cold press. The study reported a PI (peroxide index) value of 12.05 meq O_2_/kg. In our study, we collected 26 samples from six different regions—Bujaru, Bragança, Viseu, Acará, Belém, and Santarém—and found that the peroxide levels were lower compared to those of Benevides. The quality of the extracted oil can be affected by various factors, including edaphoclimatic factors like the cultivation area, harvest time, and geographic location, which have the most significant impact. However, we observed that the tree’s age had a lesser influence [53].

Out of the twenty samples tested between BT3 and BT44, eleven samples from Viseu, eight from Igarapé-Miri, and one from Bragança had a peroxide index above the acceptable limit for food purposes. This indicates that analyzing just one index is not sufficient for qualitative analysis. The suitability of a sample for food purposes depends on the results of other quality indices. Consequently, only samples BT21 to BT2D (from the municipality of Bragança), BT3B, BT3C, BT3D, BT3E, and BT3G (from the municipality of Viseu), as well as BT72 to BT74 (from the municipality of Belém), were considered suitable for food purposes according to Brazil [15] and Alimentarius [16]. It was observed that good-quality buriti oils with low levels of acidity and peroxide have potential in the regions of Bragança, Viseu, Ilha das Onças, Acará, Belém, and Santarém. In cases where buriti oil is not available for the food industry, vegetable oils with AI and PI above legislation stipulations can be used to produce bactericides, soaps, and biodiesel when adequately refined. Nanoencapsulation is an exciting alternative to oils [54,55,56]. According to a survey by Morais et al. [57], adding porcine gelatin to buriti oil increased its antioxidant potential and improved the modulation of antibiotic activity.

### 3.3. Saponification Index

The saponification indices (SI) of the oil samples are shown in Figure 3, with a population standard deviation of ±44.30, none of which exceeded the limit of 250 mg KOH/g established by legislation [15,16]. The minimum value observed was 77 mg KOH/g, referring to sample BT43 collected in the municipality of Igarapé-Miri, and the maximum was 238 mg KOH/g, quantified in sample BT71, collected in the municipality of Belém. The SI values indicate that the samples (BT21 to BT2D; BT3B, BT3C, BT3D, BT3E, and BT3G; BT72 to BT74) also fit the AI and PI parameters, according to legislation [15,16].

In a recent study by Rodrigues et al. [27], the oil extracted from buriti in the municipality of Porto Nacional in Chácara Boas Novas, Tocantins, Brazil, showed promising effects in repairing local tissue damage induced by Bothrops moojeni venom in mice. The saponification index was one of the parameters analyzed and presented a low value of 76 mg KOH/g. A similar value was observed in sample BT43 from the municipality of Igarapé-Mirí. Serra et al. [48] showed that buriti oil with an SI of 183 mg KOH/g from Benevides showed potential biological activities. Otherwise, vegetable oils with higher saponification rates may contain longer-chain fatty acids, which is desirable in many culinary applications, such as frying, as they are less susceptible to oxidation [54,58].

### 3.4. Total Carotenoid Content

Figure 4 shows that the total carotenoid (TC) content of the 50 samples, with a population standard deviation of ±412.12, varied between 308 μg/g (BT47), from the municipality of Igarapé-Miri, and 1899 μg/g (BT3B), from the municipality of Viseu. Below this value were the regions of Bragança (975 μg/g), Bujaru (771 μg/g), Ilha das Onças (716 μg/g), Belém (965 μg/g), Acará (583 μg/g), Igarapé-Miri (524 μg/g), and Santarém (492 μg/g).

According to Serra et al. [48], 1722 μg/g of TC was quantified in buriti oil in the municipality of Benevides. Also, Freitas, Ribeiro, and Nicoletti56 reported that buriti oil extracted in Ananindeua (Pará, Brazil) had good carotenoid content. The authors created optimized and stabilized emulsions with buriti oil based on the association with isolated soy protein and pectin with a high methoxyl content. The objective was an application in food products, such as dairy products, bakery products, ice creams, salad dressings, and vegetable creams. Freire et al. [59] studied the stabilization of buriti oil with whey and its transformation into lyotropic liquid crystalline structures formed during intestinal digestion, guiding digestion’s kinetics and nutrients’ bioavailability.

### 3.5. Quantification of Phenolic Compounds, Flavonoids, and Vitamin C

The values of polyphenols, flavonoids, and vitamin C in buriti oil are presented in Table 1. These compounds, along with carotenoids, are considered bioactive due to their health benefits when consumed regularly. They possess anti-inflammatory and antioxidant properties and also play a role in enhancing immune function [60,61].

The content of polyphenols in buriti oil varies from 12.04 mg GAE/100 g in Belém to 238.76 mg GAE/100 g in Igarapé-Miri. The correlation coefficient was 0.9996, indicating a strong correlation between the absorbance data and the concentration of polyphenols. The concentration of polyphenols may differ due to the maturity of the plants or seeds harvested. Generally, mature plants tend to have higher levels of polyphenols [62]. These variations may also result from the selection of raw materials, cultivation conditions, extraction and processing techniques, exposure to light, heat, and oxygen during storage, and seasonal variations [63,64,65]. According to Pannico et al. [66], the selenium levels in vegetables may influence increased polyphenols.

In Bujaru, the concentration of polyphenols was 103.05 mg of GAE/100 g, Bragança had 131.10 mg of GAE/100 g, Viseu had 109.42 mg of GAE/100 g, Igarapé-Miri had 161.68 mg of GAE/100 g, Ilha das Onças had 92.59 mg of GAE/100 g, Acará had 25.28 mg of GAE/100 g, Belém had 45.49 mg of GAE/100 g, and Santarém had 90.92 mg of GAE/100 g. It is essential to highlight that the polyphenol content in Pará is commonly quantified in buriti pulp (fruit). Therefore, no concise, updated oil data exist in the scientific literature. According to Speranza et al. [67], the concentration of polyphenols in buriti oil in Belém-PA was 107 mg GAE/Kg. In the municipality of Barreirinhas, in the state of Maranhão, the oil had a higher value of 218.02 GAE/100 g, extracted with propane [9].

This value was higher than the concentrations of polyphenols reported in previous studies of buriti oil extracted by traditional and alternative methods, which presented values below 150 mg/g [58]. Buriti fruit is known for its health benefits and has the potential for technological application and development of new products, providing antioxidant, antimicrobial, probiotic, antidiabetic, and anticancer properties [68]. Resende et al. [2] used buriti to produce flour with a high polyphenol content, making it a source of dietary fiber with high bioactive value.

The concentration of flavonoids in buriti oil was found to vary between 27.86 mg CE/100 g and 152.65 mg CE/100 g, depending on the location where it was quantified, with Bragança and Viseu having the lowest and highest concentrations, respectively. The calibration curve showed a strong linear correlation between the absorbance values and flavonoid concentrations, with a correlation coefficient of 0.9955. Bujaru, Bragança, Viseu, Igarapé-Miri, Ilha das Onças, Acará, Belém, and Santarém all had a concentration of around 100 mg CE/100 g, except for Viseu, which had a higher concentration of 125.01 mg CE/100 g.

The concentration of flavonoids in vegetable oil can be influenced by environmental and regional factors such as climate, temperature, exposure to sunlight, and soil quality. These growing conditions play a fundamental role in determining the concentration of flavonoids in buriti oil [69,70]. The extraction method used can impact the presence and quantity of bioactive compounds, such as flavonoids. Best et al. [71] conducted a study on buriti oil extracted from Peru samples using two different methods. The first was a conventional solvent-based extraction, which resulted in a content of 165.34 CE/100 g. The second method employed gas extraction under supercritical conditions, which produced a higher content of 390.82 CE/100 g.

Another study conducted by Ferreira et al. [72] found that buriti oil extracted from Piauí, Brazil, had a quercetin content of 57.91 µmol/g. These quantifications are significant due to the bioactive properties of flavonoids, which have been shown to exhibit antimicrobial, antioxidant, and anti-inflammatory activities. Additionally, flavonoids have been found to protect cells from damage caused by free radicals, thereby reducing the risk of various diseases, including cancer, cardiovascular diseases, and neurodegenerative diseases [73,74].

The concentration of vitamin C in buriti oil varies between 7.37 mg/100 g and 38.30 mg/100 g in different regions, with the municipality of Bragança showing the same range. The calibration curve demonstrated a strong linear correlation between absorbance values and ascorbic acid concentrations, with a correlation coefficient of 0.9955. In Bujaru, the concentration was 11.86 mg/100 g. In comparison, in other regions like Bragança (16.85 mg/100 g), Viseu (21.35 mg/100 g), Igarapé-Miri (24.16 mg/100 g), Ilha das Onças (16.55 mg/100 g), Acará (22.92 mg/100 g), Belém (23.50 mg/100 g), and Santarém (18.29 mg/100 g), it showed different values. It is important to note that the concentration of vitamin C in buriti oil varies depending on climatic factors and cultivation conditions, especially soil nutrition [75,76].

Although it is not expected to measure the vitamin C content of buriti oil, some studies have reported its value. For instance, a study conducted in Abaetetuba, Pará, Brazil, found that the fruit and peel samples contained 55.22 mg/100 g and 21.22 mg/100 g of vitamin C, respectively [77]. Similarly, in three regions of the Ecuadorian Amazon Forest, the average vitamin C content in the endocarp was 22.82 mg/100 g [78]. Another study conducted in Crato (Ceará, Brazil) reported a concentration of 60.63 mg/100 g of ascorbic acid in buriti pulp [79]. Additionally, Morais et al. [80] reported that buried collected in the Cerrado region contained 17.4 mg/100 g of vitamin C in the peel and 34.9 mg/100 g in flour from the dry pulp.

Fruits that contain vitamin C, such as buriti, can help strengthen the immune system, protecting against viral infections such as COVID-19 [81]. This bioactive substance is an efficient antioxidant that can reduce the risk of cardiovascular diseases and cancer [82]. In addition, it is beneficial for skin health, helping to prevent dermatological diseases, and its anti-inflammatory effect can alleviate chronic inflammatory diseases [83].

### 3.6. Fatty Acid Composition and the Application of One-Way ANOVA

Nine different types of fatty acids were detected in the collected samples. These include monounsaturated acids such as erucic, palmitoleic, and oleic acids. Although the erucic acid content is not unique to this species, changes in soil composition, climate, or agricultural practices can influence abnormalities in the fatty acid composition [84]. In addition, polyunsaturated acids such as linolenic, linoleic, and arachidonic acids were also observed, along with saturated myristic, palmitic, and stearic acids. Table 2 shows the contents of 68.41% monounsaturated acids, 19.30% saturated acids, and 12.29% polyunsaturated acids. Monounsaturated fatty acids have specific functions in the human body. Erucic acid found in vegetable oils can be used as an anti-inflammatory agent during wound healing [85]. Palmitoleic acid regulates metabolism and cell signaling [86].

Finally, oleic acid, commonly found in olive oil, is known for its cardiovascular health benefits, reducing harmful cholesterol levels and supporting artery health [87]. Polyunsaturated fatty acids are essential for regulating inflammation and the body’s immune response. They also help reduce the risk of heart disease and improve brain and eye function [88,89]. Saturated fatty acids, such as myristic, are used to synthesize some proteins [90]. Palmitic acid plays a crucial role in regulating several metabolic pathways, while stearic acid helps maintain the fluidity and integrity of cytoplasmic membranes, which is essential for proper cell function [91,92].

Oleic acid is the most abundant fatty acid found in buriti oil, comprising an average concentration of 67.34% of the total fatty acid profile. The municipality of Acará presented the highest oleic acid content, around 78%, while the municipalities of Belém and Bujaru had the lowest percentages, around 38% and 37%, respectively. These results caused a standard deviation of 17.26, indicating significant variability in fatty acid levels between these regions. In the regions of Viseu, Igarapé-Miri, Ilhas das Onças, Acará, and Belém, the levels of organic acids were below the limit of quantification (<LOQ), so the substance was blocked, but could not be quantified with isolation.

Another notable variability was observed in the linoleic acid content, with a standard deviation of 14.53, where the municipalities of Bujaru and Santarém had concentrations of 34.78% and 35.59%, respectively. The fatty acid composition of the other regions showed low standard deviations, close to zero, suggesting that the values are pretty consistent and homogeneous. A study conducted by De Souza Aquino et al. [93] analyzed buriti oil from Piauí, Brazil, and found that it contained 24.02% monounsaturated fatty acids (MUFAs), 45.01% polyunsaturated fatty acids (PUFAs), and 38.88% saturated acids (SFAs).

Another study by Anjos et al. [9], conducted in Barreirinhas, Maranhão, Brazil, extracted buriti oil with pressurized propane and found that it contained 19.45% SFA, 77.44% MUFA, and 3.11% PUFA. Meanwhile, a study by Serra et al. [48] analyzed cold-pressed buriti oils from Benevides and found that they contained 19.98% SFA, 78.81% MUFA, and 1.22% PUFA. These studies show that the composition of fatty acids in vegetable oils is influenced by environmental conditions such as climate, soil, and sunlight levels, as well as by the genetic characteristics of the species in each region [86,94,95]. One-way ANOVA analysis was performed in eight regions of Pará, resulting in a *p*-value less than 0.05 and a 95% confidence level.

Using Fisher’s method, the average between groups represented by the regions in the graph was 11.11% (Figure 5). The analysis revealed no significant differences between the groups in terms of fatty acid content, indicating that the means of all groups are statistically similar under the study conditions and based on the data collected.

The interpretative analysis showed that the regions of Bragança (BT2), Viseu (BT3), Igarapé-Miri (BT4), Ilha das Onças (BT5), Acará (BT6), and Belém (BT7) had the highest levels of oleic acid, with an average of 77.31%. The absence of outliers in the Bujarú (BT1) and Santarém (BT8) regions indicated low acid levels, 36.96% and 37.94%, respectively. It is essential to note that the conclusion regarding the equality of means between groups does not imply that the different fatty acid contents within each group are numerically identical. Instead, any difference observed is considered statistically insignificant within the study’s limits and the data’s variability. The ANOVA analysis has revealed that the fatty acid composition of vegetable oils can vary significantly depending on the region where they are produced. These variations can be attributed to various factors such as the soil type, climate, growing conditions, and processing practices. Several studies [96] have highlighted the importance of these findings in fine-tuning the production processes and ensuring consistent oil quality across different locations.

### 3.7. Principal Component Analysis and Quality Control

This study analyzed various parameters such as AI, PI, SI, carotenoids, polyphenols, flavonoids, and vitamin C to determine their contribution to the quality of buriti oil. This study used Principal Component Analysis (PCA) to identify the most significant variables that determine the quality of buriti oil. The study also detected two anomalous observations in samples BT1 and BT36 using the Mahalanobis distance metric (Figure 6). Based on the data variance, this metric calculates the distance between a reference point and a given point. The research cited studies by Vishwakarma et al. [97], Jainalabidin et al. [98], and Jin et al. [99] to explain the use of the Mahalanobis distance metric.

In the first screening, an unusual observation above 3.9 was detected. Sample BT1 was identified and removed. As it had the highest acidity value, with a value of 19.2 mg of KOH/g, the model indicated that the sample was not of good quality. In the second screening, sample BT36 was identified and removed from the model at a position above 3.9, as it was considered an unusual observation, possibly due to the high quantified peroxide index of 15.6 meq O_2_/kg.

Finally, in the last screening, there were no unusual observations; all samples clustered below 3.9. In quality control applications, the Mahalanobis distance can be used to evaluate the quality of products or production batches, so very high values may indicate vegetable oils that are out of specifications or adulterated [100,101,102]. The Principal Component Analysis (PCA) revealed that the first two components accounted for an accumulated variance of 98.70%. The first principal component (PC1) explained 96.30% of the total variance, and the second principal component (PC2) accounted for 2.4%. The model (Figure 7) was designed to analyze the quality of buriti oil based on its parameters. The order of importance of the parameters was established using the eigenvectors calculated in the main components. In PC1, the order of decreasing importance was as follows: carotenoids, flavonoids, PI, polyphenols, AI, vitamin C, and SI, with respective loadings of 0.999, 0.037, 0.0011, 0.001, −0.0001, −0.003, and −0.016. In PC2, the order of importance was specified as follows: SI, carotenoids, AI, vitamin C, PI, flavonoids, and polyphenols, with respective loadings of 0.252, 0.009, 0.005, −0.004, −0.029, −0.088, and −0.963.

PCA has been used in previous studies to identify the critical components in vegetable oils that explain most of the variability in the data. This can help to determine the key characteristics that affect oil quality. In a recent Ndiaye et al. [103] study, only three batches of Adansonia digitata seed oil were subjected to three different temperatures. Subsequently, they analyzed different quality indices and found that 80.48% of the variance accumulated from the eigenvectors distributed in the two main components. In the research by Serra et al. [48], although the PCA used had four main components, the accumulated variance was 87.7%. This helped differentiate the oil samples (pracaxi, patauá, passion fruit, Brazil nut, and buriti) and fat (ucuúba, cupuaçu, muru-muru, and bacuri). This modeling made it possible to verify the quality of these plant matrices based on data on acidity, peroxide, saponification, total carotenoids, and other parameters.

Principal Component Analysis (PCA) was used to manage a model that identified three groups based on biological activity (Figure 8), using carotenoid content as the primary parameter. Carotenoids have a complex and multifaceted relationship with biological activity, which depends on factors like the type of carotenoid, the food matrix, individual factors, and analysis methods. Generally, a sample with higher carotenoid content may have more excellent biological activity, especially concerning its antioxidant potential [104]. However, the relationship between the two factors can vary based on several other factors, such as the type of carotenoid present, the compound’s bioavailability, interactions with other nutrients, and the biological context in which they are acting according to studies by Bendjabeur et al. [105] and Zhang et al. [106].

The loadings plotted in the PC graph after dimensionality reduction and reporting that carotenoids represent the most relevant parameter in sample quality, three regions were perceived to justify the potential for biological activity. The samples in the first group, represented by a blue rectangle, showed the lowest potential for biological activity, in PC1 scores that ranged from 0.028 to 0.9881. The second group, represented by a yellow rectangle, had intermediate values in PC1 scores ranging from 0.1017 to 0.1491. The third group, represented by a red rectangle, had the highest biological activity values in PC1 scores that ranged from 0.1543 to 0.2703.

In the municipality of Bujaru, all samples were categorized under the low biological potential (LOP) group. In Bragança, one sample (7.14%) was identified in the LOP group, another in the medium biological potential (MEP) group, and 85.72% of the samples were grouped in the high biological potential (HIP) category. In Viseu, 6% of the samples were in LOP, 12% in MEP, and 82% in HIP. Igarapé-Miri had 25% of the samples in MEP and 75% in LOP. The only sample from the Onças Islands was placed in the MEP group. All three Acará samples remained in the LOP group. In Belém, 60% of the samples were categorized under MEP and 40% under LOP. The only sample from Santarém was grouped in the LOP category.

In a study by Zhang et al. [49], Principal Component Analysis (PCA) was utilized to evaluate the quality indices of four hotpot oil formulations. These formulations were a mixture of beef tallow, lamb tallow, peanut oil, and palm oil. According to the authors, when the total variance rate exceeds 85%, the identified principal components can be considered as reflective of all factors. Similarly, they used PCA to obtain a general description of the sample distribution and possible grouping into a homogeneous data group [107].

In the municipality of Bujaru, all samples were categorized in the low biological potential (LOP) group. In Bragança, one sample (7.14%) was identified in the LOP group, another in the medium biological potential (MEP) group, and 85.72% of the samples were grouped in the high biological potential (HIP) category. In Viseu, 6% of the samples were in LOP, 12% in MEP, and 82% in HIP. Igarapé-Miri had 25% of the samples in MEP and 75% in LOP. The only sample from Ilhas das Onças was placed in the MEP group. All three samples from Acará remained in the LOP group. In Belém, 60% of the samples were categorized as MEP and 40% as LOP. The only sample from Santarém was grouped in the LOP category.

They found that applying silver and copper nanocolloids concentrated at 120 mg/dm^3^ increased carotenoid levels, intensifying biological activity. The total variance reached 85%. Thus, compared to these studies, our model presented three groups of biological activity stages that estimated 13.70% more total variance, which can be characterized as a more robust model. PCA can be a useful tool in vegetable oil quality control. It can efficiently analyze biological activity and other quality parameters when applied correctly. In addition, outliers should be identified, and quality thresholds based on the eigenvector groups, arranged according to the position of the oil samples in the principal component scoring space, should be established. This information can assist in qualifying the samples [108,109,110].

## 4. Conclusions

The physical–chemical analyses of the quality indices established in the legislation identified that nine samples of buriti oils would fit into the food sector and the others into other sectors, depending on the processing appropriate to the needs of different uses. Among the four bioactive compounds, total carotenoids presented the highest concentration. Monounsaturated fatty acids were predominant in buriti oils, especially oleic acid, the majority in all scored samples. The PCA identified the order of relevance of the relative quality profile parameters and segregated three distinct groups of samples based on biological activity. One-way ANOVA showed no significant differences between regions about the content of fatty acids stipulated in considerable ranges. However, it justified differences within five regions with oleic acid levels, primarily expressed in their fatty content, which can infer more excellent added value for these oils. Although chemometric methods are currently functional and based on a specific data set over a set period, there is room for refinement and optimization in future research. These improvements can be achieved by increasing the sample space to contribute to various economic sectors, such as the food and agricultural industries. This approach to this study has the potential to identify better-performing cultivation areas and monitor the authenticity of this oilseed matrix.

## Figures and Tables

**Figure 1 foods-14-01585-f001:**
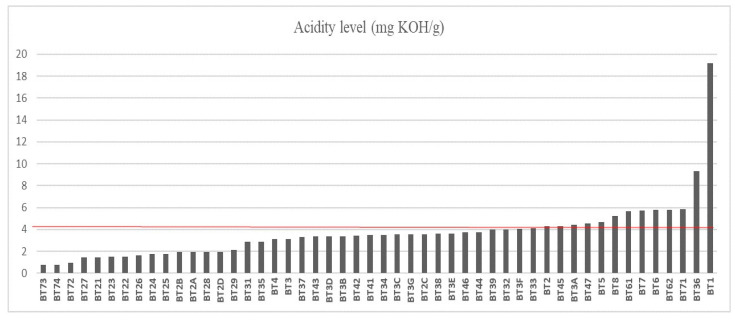
Acidity index of buriti oil. The line highlighted in red represents the limit of fatty acid concentration according to oil quality criteria.

**Figure 2 foods-14-01585-f002:**
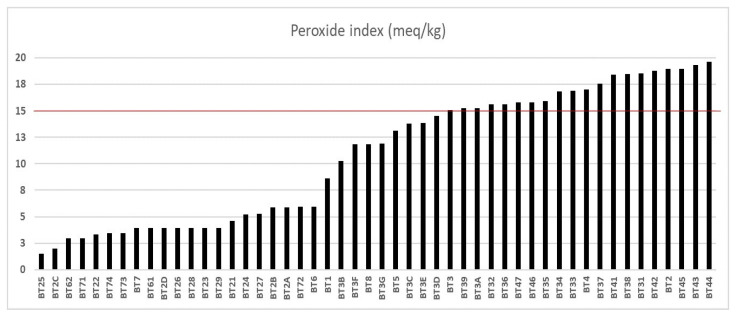
Peroxide index of buriti oil. The line highlighted in red represents the limit of meq O_2_/kg concentration according to oil quality criteria.

**Figure 3 foods-14-01585-f003:**
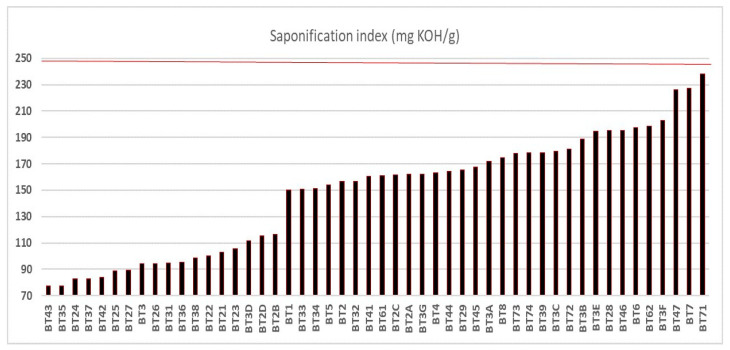
Saponification index of buriti oil. The line highlighted in red represents the limit of mg KOH/kg concentration according to oil quality criteria.

**Figure 4 foods-14-01585-f004:**
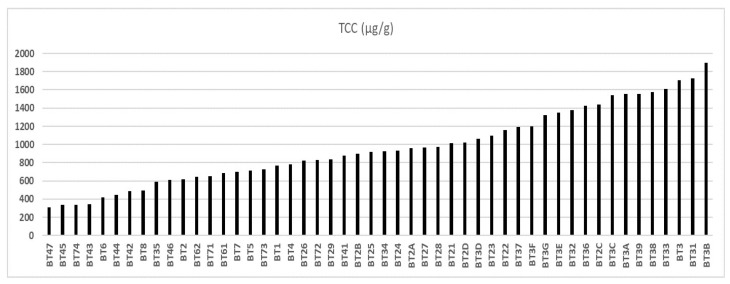
Total carotenoid content of buriti oil.

**Figure 5 foods-14-01585-f005:**
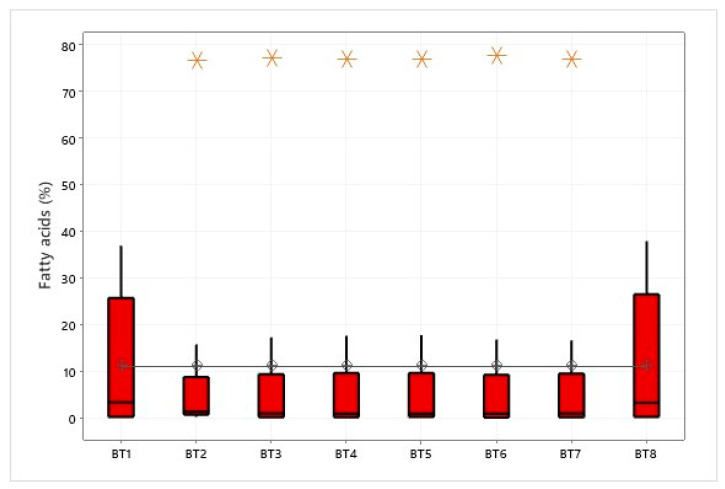
One-way ANOVA for fatty acids in different regions of Pará, Brazil.

**Figure 6 foods-14-01585-f006:**
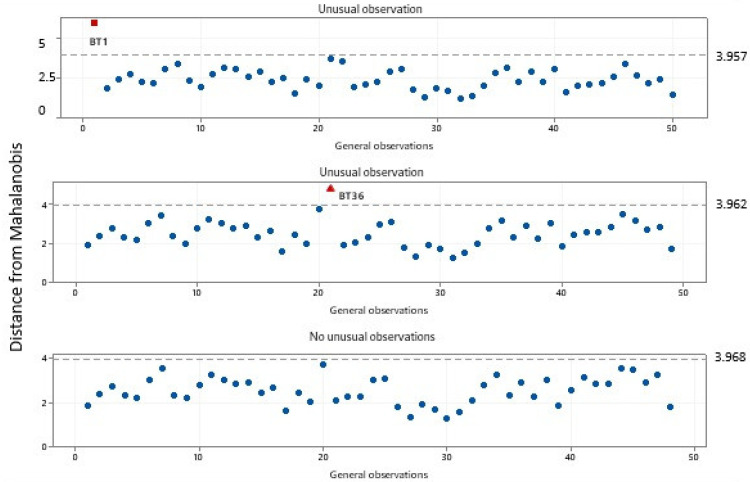
Outliers observed by the Mahalanobis distance (highlighted in red) after analyzing the quality parameters.

**Figure 7 foods-14-01585-f007:**
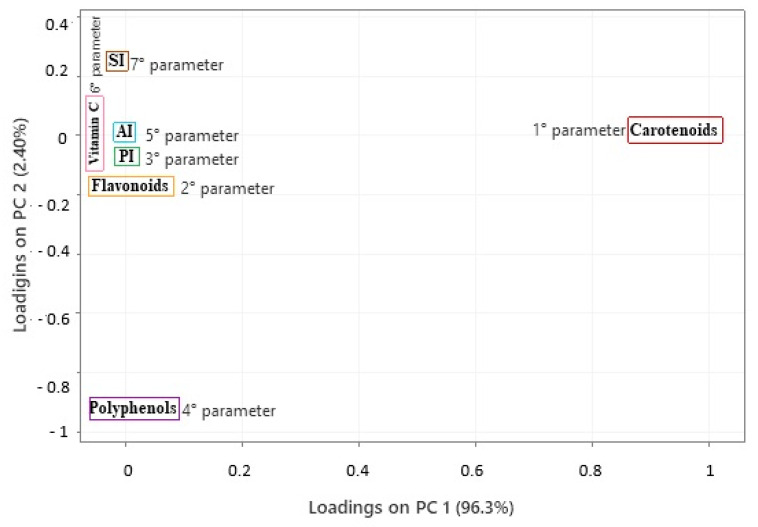
Projection of quality parameters, carotenoids, flavonoids, polyphenols, and vitamin C of buriti oil samples in Pará (Brazil) locations. PI: peroxide index; SI: saponification index; AI: acid index.

**Figure 8 foods-14-01585-f008:**
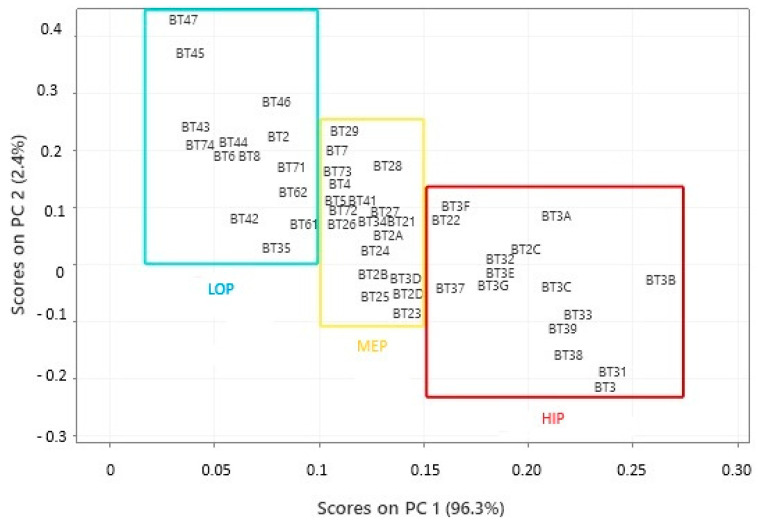
Projection of buriti oil samples in different locations in Pará (Brasil) based on the potential for biological activity.

**Table 1 foods-14-01585-t001:** Values of polyphenols, flavonoids, and vitamin C in buriti oil.

Samples	Flavonoids (CE/100 g)	Polyphenols (GAE/100 g)	Vitamin C (mg/100 g)
BT1	108.24	103.06	11.87
BT2	93.17	172.68	18.50
BT21	103.63	184.94	10.56
BT22	91.92	215.32	13.87
BT23	114.20	70.04	29.29
BT24	126.51	140.18	18.71
BT25	134.28	49.15	15.61
BT26	117.83	155.95	38.31
BT27	115.08	206.75	20.42
BT28	85.04	140.90	13.58
BT29	121.15	210.62	12.64
BT2A	27.86	69.19	13.13
BT2B	44.10	36.76	11.14
BT2C	99.75	147.67	7.37
BT2D	48.29	35.36	12.89
BT3	128.72	85.56	26.86
BT31	133.52	103.14	29.09
BT32	132.76	107.46	25.83
BT33	152.65	87.89	14.36
BT34	134.71	108.66	17.23
BT35	126.69	75.85	10.33
BT36	124.99	131.69	35.02
BT37	122.61	152.52	29.10
BT38	123.32	107.16	25.12
BT39	132.52	39.10	19.30
BT3A	141.49	237.41	12.41
BT3B	120.29	184.34	19.03
BT3C	118.69	100.39	16.36
BT3D	113.50	96.34	17.94
BT3E	110.03	60.24	17.95
BT3F	100.79	106.51	24.01
BT3G	108.07	76.03	23.10
BT4	88.15	129.14	23.45
BT41	103.02	92.40	14.79
BT42	70.18	100.05	28.95
BT43	79.98	233.20	28.48
BT44	88.16	117.95	29.82
BT45	89.04	238.76	28.27
BT46	77.04	172.79	14.00
BT47	73.71	209.22	25.57
BT5	117.58	92.59	16.55
BT6	73.72	28.96	18.85
BT61	74.43	26.08	26.00
BT62	79.87	20.80	23.94
BT7	53.54	12.04	15.02
BT71	126.80	18.45	23.69
BT72	133.43	59.85	25.44
BT73	71.56	83.52	27.09
BT74	70.98	53.64	26.27
BT8	77.36	90.93	18.30
** *Me* **	102.10	111.58	20.51
** *Sd* **	27.82	61.91	6.99

**Table 2 foods-14-01585-t002:** Fatty acid profile of collected samples. Relative concentration (%).

Samples	Myristic	Palmitic	Palmitoleic	Stearic	Oleic	Linoleic	Linolenic	Arachidic	Erucic
BT1	0.16	16.71	0.12	3.05	36.96	34.79	3.34	0.26	4.59
BT2	1.80	15.81	0.19	1.57	76.94	1.36	0.91	0.90	0.50
BT3	0.04	17.29	0.20	1.52	77.49	1.52	0.99	0.81	<LOQ
BT4	0.05	17.65	0.23	1.49	77.22	1.61	0.97	0.68	<LOQ
BT5	0.04	17.77	0.20	1.50	77.00	1.50	0.92	0.86	<LOQ
BT6	0.04	16.85	0.18	1.68	77.99	1.46	0.94	0.82	<LOQ
BT7	0.05	16.63	0.21	1.70	77.22	2.38	1.00	0.71	<LOQ
BT8	0.17	17.49	0.16	3.29	37.94	3.60	3.71	0.25	1.40
*Me*	0.30	17.02	0.19	1.97	67.34	10.03	1.60	0.66	0.81
*sd*	0.57	0.61	0.03	0.69	17.26	14.53	1.12	0.24	1.50

## Data Availability

The original contributions presented in the study are included in the article, further inquiries can be directed to the corresponding author.

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
