# Peer review of "Chemometric Tools Associated with Quality Parameters for Evaluation of Mauritia flexuosa L.f. Oil in the State of Pará (Brazil)"

_foods, 2025, doi:10.3390/foods14091585_

Round 1
Reviewer 1 Report
Comments and Suggestions for Authors
Please check language errors and condense manuscript content throughout the entire text.
Author Response
Dear reviewer,
Please find attached the answers point by point.
Thank you in advance for your considerations.
Best regards,

Reviewer 2 Report
Comments and Suggestions for Authors
Chemometrics associated with quality and fatty acid composition parameters for evaluation of Buriti Oil (Mauritia flexuosa) in the State of Pará (Brazil)
The overall goal of this study is to evaluate the quality of buriti oil in 50 samples coming from Para, Brazil. Based on the abstract, the study aims to determine the quality of buriti oil based on samples provided. The abstract is unclear. I believe there are also spelling errors in some words in the manuscript. For example, in the abstract, I believe this should be ‘carotenoids’ and not ‘carotnoids.’ I understand the main goal based on the abstract but there seems to be incoherence on how this goal will be achieved. For example, how is determination of various parameters (e.g., the levels of total carotenoids, polyphenols, flavonoids, vitamin C and antioxidant activity) related the overall goal and PCA analysis. In general, the abstract was not written clearly although I think the results are interesting. It is challenging to appreciate the overall intent and significance of the project based on the abstract.
In the introduction, please provide more benefits of the buriti oil as well as its abundance, source, and economic demand.
Please define the acronym and only use that acronym all throughout the manuscript. An exception is in the abstract.
It is unclear as to what is the significance of assessing the quality of buriti oil and how it is done based on the introduction. This should be substantiated.
What were the previous or any related studies done? What is the novelty of the study?
There appears to be variability in the experimental results. For example, some values for the acid index as well as peroxide index appear to be acceptable limits for some samples but not for the others. It all depends on the sources as mentioned in the paper. Although there are some explanations as mentioned in the manuscript leading to such results, they appear to be not convincing or certain. More elaborate explanations are needed.
Overall, the study is interesting but I think the study lacks novelty.
Comments on the Quality of English LanguageThe manuscript requires moderate English language copyediting.
Author Response

(The authors gave the same response as above.)
